# Emotional Contagion Among Adolescents with Type 1 Diabetes Mellitus (T1DM) and Their Primary Caregivers: Benefits of Psychological Support for Family Systems in Pilot Study

**DOI:** 10.3390/children12020151

**Published:** 2025-01-28

**Authors:** Pilar Rodríguez-Rubio, Javier Martín-Ávila, Esther Rodríguez-Jiménez, Selene Valero-Moreno, Inmaculada Montoya-Castilla, Marián Pérez-Marín

**Affiliations:** Department of Personality, Assessment and Psychological Treatments, Faculty of Psychology and Speech Therapy, University of Valencia, 46100 Valencia, Spain; mpiroru@alumni.uv.es (P.R.-R.); javier.martin@uv.es (J.M.-Á.); esther.rodriguez-jimenez@uv.es (E.R.-J.); selene.valero@uv.es (S.V.-M.); inmaculada.montoya@uv.es (I.M.-C.)

**Keywords:** diabetes, emotional contagion, adolescents, family caregiver, stress, emotional distress, psychological therapy

## Abstract

Background. T1DM is a significant chronic condition that necessitates regular medical monitoring, dietary and physical activity supervision, and daily blood glucose monitoring and insulin therapy. The management of this disease and the transition to adolescence often have a significant psychosocial impact on the individual and their family. Objective. The objective of this correlational study was to examine the reciprocal influence between adolescents and their caregivers, with a particular focus on the beneficial effect of receiving psychological support sessions from family members and adolescents with T1DM in a pilot study. Methods. An indicator variable was developed to facilitate an analysis of changes occurring prior to, as well as following, the administration of the treatment in question. Family caregivers received two therapy sessions, and we analyzed their perceived caregiver stress. Adolescents had five sessions, and the reduction in emotional distress was studied in them. Results. The sample comprised 15 adolescent–family caregiver dyads. All parents were mothers of adolescents, with a mean age of 47.67 and 13.47 years, respectively. Descriptive statistics and Spearman correlations were conducted. Following the completion of the psychological counseling sessions, the data revealed a significant positive correlation between the perceived reduction in global stress experienced by the caregiver and the adolescent’s emotional distress, with correlation coefficients of 0.74 and 0.61, respectively. Furthermore, a positive relationship was observed between the reduction in existing difficulties in family role adjustment and the reduction in emotional distress among diabetic youth, with correlation coefficients of 0.72 and 0.57. The frequency of emotional distress of the caregiver also correlated with adolescent emotional distress, with a coefficient of 0.60. Conclusions. The findings of this study provide evidence for the circularity of family systems change. A positive emotional contagion effect is observed in the improvements in stress and emotional distress experienced during adolescence and in the family’s adjustment to T1DM, as reported by caregivers and their children who received psychological support sessions.

## 1. Introduction

Chronic diseases (CD) have a global impact, resulting in 41 million deaths per year (74% of deaths in total). The most prevalent of these are cardiovascular diseases, chronic respiratory diseases, cancer, and diabetes [1]. Despite the progressive increase in its global incidence over time [2], type 1 diabetes mellitus (T1DM) typically manifests during the infantile–juvenile stage [3,4]. It is the most prevalent disease in this age group, exhibiting two incidence peaks: one between the ages of 5 and 9, and another between the ages of 10 and 14 [5]. The treatment of this disease involves the continuous monitoring of blood glucose levels, administration of insulin, adherence to a diet, and consistent physical exercise [6,7]. Noncompliance with these guidelines can result in various complications, including hypoglycemia or hyperglycemia, as well as other serious health concerns that may even lead to mortality [8,9].

During adolescence, living with a chronic condition such as T1DM can present significant physical and psychosocial challenges for both the adolescent and their family [10]. Research indicates that adolescents often exhibit suboptimal glycemic control in comparison to other age groups [11,12]. Given that parents typically serve as the primary informal caregivers, their responsibilities can result in elevated levels of diabetes-related burden and distress [13], as well as psychological problems, such as depression [14,15]. The impact of living with this disease at the family level forces family members to adapt and find a balance between their lifestyles and the needs of the adolescent with T1DM [16]. This significant change can sometimes result in family conflicts, which have been associated with lower adherence and higher HbA1c levels [17,18]. Likewise, the presence of psychosocial problems at this age may negatively affect adequate adjustment to the disease, adherence to treatment, and glycemic control [19,20]. Conversely, evidence suggests that positive family functioning and cohesion promote the distribution of disease-related tasks, enhancing glycemic control and self-management among adolescents [21,22,23]. These family variables appear to enhance resilience in children, which may in turn influence better disease management outcomes [24]. In general, the family is perceived as an indispensable support for these adolescents, whose presence can improve their emotional well-being, self-efficacy, and self-management of T1DM [25].

The extant scientific literature on the subject suggests the presence of emotional contagion between primary caregivers and patients with chronic diseases, finding an association between the presence of such diseases in a family member and higher levels of depression and anxiety among other family members [26]. Studies in other chronic diseases have also shown a relationship between the anxious–depressive symptomatology of both caregivers and adolescents, although the results remain inconclusive [27,28]. While it has been observed that the disease significantly affects both parties [29,30], the direct interaction between the two is not fully understood, as it depends on several factors such as the severity of the disease, the duration and coping mechanisms of the individual or changes in family routines as discussed above [22,24]. Some research highlights that the severity of the illness plays a crucial role in determining the extent to which psychological symptoms manifest themselves in caregivers and adolescents. In addition, other extrinsic factors, such as familial support, perceived threat of illness, and access to healthcare, may also influence the emotional outcomes of both caregivers and adolescents [30]. Due to the intricacy of these interrelationships, further studies are necessary to explore how these factors interact over time and to gain a more nuanced understanding of the emotional dynamics between caregivers and adolescents in chronic illness contexts.

In light of the aforementioned points, this present study endeavors to examine the merits of the implementation of a psycho–socio–emotional intervention program, designated 10Vida, for adolescents diagnosed with T1DM and their primary caregivers. The focal point of this study is the emotional contagion that transpires within family systems grappling with a chronic condition of this nature.

The 10Vida treatment program was developed to intervene at a psycho–socio–emotional and educational level in adolescents with T1DM and their families in a hospital setting, providing them with the tools necessary to adequately adjust to and manage the condition. The program comprises seven sessions, with a total duration of six months, with the first five sessions targeting adolescents and the subsequent two sessions focusing on parents. The adolescent sessions are designed to enhance self-esteem, emotional regulation, and social relationships, while the caregiver sessions emphasize stress management and the establishment of family support systems [31].

This present study seeks to analyze the benefits of the 10Vida psycho–socio–emotional intervention program, as applied to adolescents with T1DM and their primary caregivers. This study will focus on the emotional contagion that occurs during this stage in family systems affected by a chronic disease such as this one. In relation to the objective, the following hypotheses are proposed: (1) perceived caregiver stress will be positively associated with the emotional symptomatology of the adolescent with T1DM, and vice versa. (2) Exploratory analyses suggest that completion of the 10Life program will result in lower scores on emotional symptomatology and stress levels in both caregivers and adolescents with T1DM.

## 2. Materials and Methods

### 2.1. Sample Description

The sample comprised 15 primary family/caregiver–adolescent dyads with T1DM who consistently attended the pediatric endocrinology service of the Hospital General Universitario in Valencia, Spain, for follow-up. In order to be considered a participant in this study, the following criteria had to be met: (a) a minimum age of 12 years, (b) a minimum duration of diagnosis with T1DM of six months, and (c) regular attendance at the pediatric endocrinology outpatient department of a hospital for medical follow-up. Patients with the following conditions were excluded from participation: (a) infantile cerebral palsy (ICP) or epilepsy, (b) brain tumors, (c) attention deficit hyperactivity disorder (ADHD), and (d) those with a psychological diagnosis prior to the onset of the diagnosis of organic disease.

### 2.2. Design and Procedure

A preliminary assessment of the pertinent variables was conducted on a sample of 15 caregiver–adolescent dyads prior to the implementation of the 10Vida intervention program, which spanned a duration of 6 months and comprised five therapeutic sessions designed for the adolescents (S1–S5 Adolescents in Table 1) and two sessions targeting their caregivers (S1 and S2 Parents in Table 1). The specific details of these sessions are outlined in Table 1 below. The fieldwork, evaluation, and intervention were carried out entirely by one of the team’s researchers, an expert in health psychology who had received training in the cognitive–behavioral therapy model.

Following the completion of the 10Vida program by the participating dyads, a second data collection was conducted using the questionnaires and tools outlined in the subsequent section. Consequently, the participants’ results were recorded at two distinct points in time (pre- and post-intervention). This study commenced in 2021 and concluded in the first quarter of 2024. Prior to participation, all participants were informed of the study methodology and provided their prior consent to take part in the study. Any modifications to the protocol were documented in ClinicalTrials.gov (NCT04476433). This study was conducted in accordance with the ethical guidelines established in the 2013 World Medical Association Declaration of Helsinki and was approved by the Human Research Ethics Committee of the Universitat de València (Reference: 1226194) and the Hospital General Universitario de Valencia (Reference: 151/2021).

Appropriate measures were implemented to ensure the complete confidentiality of the participants’ data, in accordance with the Organic Law on the Protection of Personal Data (LOPD) 3/2018, of 5 December.

### 2.3. Variables Analyzed

A computerized questionnaire was developed using the measurement instruments described below and implemented through LimeSurvey, an online evaluation platform provided by the Universitat de València.

#### 2.3.1. Sociodemographic and Clinical Variables

An ad hoc registry was utilized to collect the sociodemographic variables of gender and age for both adolescents and caregivers in the sample. In addition, clinical variables were documented, including secondary diagnoses, the duration of treatment with T1DM, the frequency of hospital admissions due to T1DM, the duration of these admissions, adrenaline use, and the frequency of regular medical control visits. Caregivers’ information will be collected as well, including their employment status, duration of employment, marital status, socioeconomic level, relationship to the adolescent with T1DM, and level of education.

#### 2.3.2. Psychological Variables

##### Psychological Variables of Adolescents

Strengths and Difficulties Questionnaire (SDQ) [32]. The SDQ, developed by Goodman et al. (2001), is a tool employed to evaluate the potential presence of psychopathology and emotional adjustment in adolescents with type 1 diabetes mellitus (T1DM). The primary objective of the SDQ is to identify emotional or behavioral disorders in children and adolescents between the ages of 4 and 16. The instrument comprises 25 items, which are structured in a Likert-type format with three response options (0: Not true; 2: Truly yes). The items are grouped into five subscales: emotional symptomatology, behavioral problems, hyperactivity, peer problems, and prosocial behavior. In addition to the specific scores for each subscale, an overall questionnaire score can be calculated by summing the subscales, with the exception of “prosocial behavior”. The SDQ has been widely used in previous studies and has shown acceptable reliability in various populations, including those with chronic diseases. For instance, in the study by Valero-Moreno et al. [33] on patients with chronic respiratory disease, reliability coefficients of 0.72 for the total score, 0.61 for emotional symptoms, 0.43 for behavioral problems, 0.63 for hyperactivity, 0.49 for prosocial behavior, and 0.61 for peer problems were reported. To assess the potential psychopathology and emotional adjustment of adolescents with T1DM, this questionnaire is employed. The primary objective of this instrument is to detect possible emotional or behavioral disorders in children and adolescents aged 4 to 16 years.

Hospital Anxiety and Depression Scale (HADS) [34]. This instrument, originally developed by Zigmond and Snaith [34], was used to assess the cognitive clinical features of anxiety and depression in the adolescent sample [34]. In adolescents, the version adapted and validated by Valero-Moreno et al. [35] will be used. This version is composed of 11 Likert-type items (instead of the original 14), with a four-point scale ranging from 1 (strongly agree) to 4 (strongly disagree). As in the original version, it is composed of two subscales with the dimensions “depression” and “anxiety”, which together constitute the global factor of emotional distress that was used in this research. Regarding the internal consistency of the scale, it has shown reliability values of 0.86 for the anxiety scale, 0.78 for the depression scale, and 0.88 for general emotional distress [33].

##### Psychological Variables of Caregivers

Pediatric Inventory for Parents (PIP) [36]. The present questionnaire was developed to evaluate the stress experienced by caregivers when confronted with the demands associated with caring for adolescents with type 1 diabetes mellitus (T1DM). In this case, the Spanish translation provided by Del Rincón et al. [37] was utilized. The original instrument comprises 42 items that delineate potentially stressful situations in the hospital context for parents with chronically ill children. Each item is evaluated according to its frequency and the effort involved, yielding two overall scores that are subsequently distributed into different dimensions. The questionnaire employs a Likert-type scale ranging from 1 (not at all) to 5 (very much), with all items formulated directly. Additionally, it is structured into four specific subscales: “Communication”, “Medical care”, “Emotional functioning”, and “Family role” [37]. The questionnaire also comprises two global total scores that assess the perceived stress generated by the frequency of caregiving tasks and the perceived stress caused by the effort involved in caregiving tasks. According to the original study by Streisand et al. [36], Cronbach’s alpha coefficients obtained for each subscale are as follows: in the “Communication” dimension, the alpha was 0.89 for the Frequency subscale and 0.90 for the Effort subscale; in “Medical care”, values of 0.82 and 0.88, respectively, were reported. The “Emotional Functioning” dimension exhibited an alpha of 0.83 for the Frequency subscale and 0.90 for the Effort subscale. In the “Family/Social Role” dimension, the coefficients were 0.79 for the Frequency subscale and 0.85 for the Effort subscale. This instrument serves as a valuable instrument for the analysis of the demands perceived by caregivers and their impact on various domains of care for adolescents with T1DM.

### 2.4. Statistical Analysis

All statistical analyses were executed using the statistical software “SPSS Statistics”, version 28.0.1.1, on the Windows 11 operating system. In order to respond to the hypotheses mentioned, it was necessary to perform an analysis of the most relevant descriptive statistics (frequency, mean, median, standard deviations, minimum, and maximum). Likewise, Spearman correlations were employed to examine the relationships between the study variables—perceived stress, caregiver subscales, and the emotional distress and difficulties of the adolescent—given that some variables did not meet normality assumptions and considering the sample size.

## 3. Results

### 3.1. Sociodemographic and Psychological Variables

Table 2 presents the primary findings concerning the clinical and sociodemographic characteristics of the caregivers. A salient point is that the totality of the sample comprised exclusively women (100%), with 14 of them (93.3%) concurrently engaged in employment and serving as primary caregivers for adolescents with T1DM. The majority of caregivers (86.7%) were either married or cohabited with a partner, while a smaller percentage was separated (6.7%) or divorced (6.7%). It is noteworthy that 33.3% of the caregivers reported experiencing at least one physical health problem, and 13.3% indicated that they had a psychological health problem.

Regarding the kinship relationship of the caregiver to the adolescent with T1DM, 94.7% were their mothers, and 5.3% (only one case) were their grandmothers.

A consideration of the case of the adolescents reveals that the sample was composed of nine males (60%) and six females (40%). Of these, four had a physical disease in addition to T1DM (26.7%) and two had a psychological disorder (13.3%). The most common frequency of visits was every 3 months (73.3%). This data can be found in Table 3.

Finally, Table 4 shows the results of the remaining sociodemographic variables. The mean age of the subjects was 13.46 years (SD = 1.68), with a minimum age of 11 (one subject turned 12 during the course of the study) and a maximum age of 17. The mean duration of treatment for T1DM was 45.26 months (SD = 40.24, Md = 33), with a minimum range of 6 months and a maximum range of 138 months. The frequency of hospital admissions due to T1DM was low (M = 1.60, SD = 1.639, MD = 1), and the duration of these admissions was even lower (M = 1.20, SD = 1.74, MD = 0). The mean age of the caregivers was 47.66 years (SD = 7.21), with a minimum of 35 and a maximum of 65 years. Those who were active (n = 14) had been working for a mean of 142.31 months (SD = 132), with a minimum of 12 and a maximum of 276 months.

### 3.2. Psychological Variables

#### 3.2.1. Profile of Pre-Intervention and Post-Intervention Variables

As illustrated in Table 5, the psychological scores obtained prior to the intervention revealed a medium–high profile of perceived stress due to the frequency of caregiving tasks (PIP). The mean score of 10.18 (SD = 2.15, MD = 9.75) and the maximum range of this test (15) indicate a medium–high level of stress.

Conversely, the scale of perceived stress due to the effort involved in caregiving tasks exhibited low mean scores (M = 7.35; SD = 3.20), indicating that caregivers perceive greater stress stemming from the frequency of caregiving tasks they provide to adolescents with T1DM as opposed to the effort involved in these caregiving tasks. Conversely, the mean scores obtained in emotional distress from both the HADS questionnaire and the SDQ did not demonstrate the presence of significant clinical signs in this regard. The mean emotional distress score according to the HADS (M = 6.20; SD = 5.45) is below 20, which is the cut-off score indicating relevant emotional distress. The mean of significant emotional symptoms according to the SDQ is 10.13 (SD = 6.84), which is within the normal range.

A subsequent examination of the post-intervention variables reveals no substantial alterations; however, a general decline in the means of all the variables under scrutiny is evident. The most pronounced decrease is observed in the variable representing perceived general stress associated with the effort demanded by caregiving responsibilities, which diminishes from an average of 7.35 to 6.06. The variable that exhibited the least changes was that of the adolescent’s emotional discomfort, which decreased from an average of 6.20 to 5.93. However, if we examine the median of this variable, we observe a decrease from 5 to 4.

#### 3.2.2. Variable Difference Between Pre- and Post-Intervention

As illustrated in Table 5, the mean of the differences between pre-intervention and post-intervention scores was calculated, thereby obtaining a measure of the variation in scores among time points. The sign of these variations indicates that the change of the scores after the intervention was generally positive since the mean pre-intervention scores for both parental stress and adolescent distress and emotional adjustment were higher than the post-intervention scores.

#### 3.2.3. Correlations Between the Variation of the Different Scores

The correlation matrix presented in Table 6 explores the relationships between variations in stress due to caregiving tasks and general adolescent emotional well-being outcomes. Several significant correlations were identified, offering insights into the interconnectedness of caregiving stress and adolescent emotional health. We observed a significant correlation (ρ = 0.721) between the variation in caregivers’ perceived stress by the frequency of caregiving tasks related to the family role and emotional distress in adolescents with T1DM, as measured by the HADS.

Additionally, a significant correlation (ρ = 0.572) was observed between the variations in caregivers’ perceived stress due to the frequency of caregiving tasks related to the family role and general adolescent emotional adjustment. This variable also presented a positive significant correlation with the variation of emotional distress frequency perceived by the caregivers (ρ = 0.602). We also observed a significant correlation between the variation in perceived stress due to the overall task frequency of the caregivers of these adolescents, with the emotional distress reported by the adolescents (ρ = 0.740) and the presence of emotional psychopathology, as measured by SDQ (ρ = 0.614). These results also point to a lack of significant correlation between the variation in perceived stress stemming from the effort associated with caregiving tasks and the emotional distress or adjustment of the adolescents.

## 4. Discussion

The primary aim of this study was to analyze the phenomenon of emotional contagion within the dyad of adolescents with type 1 diabetes mellitus (T1DM) and their caregivers. Our findings align with the existing literature indicating that a significant proportion of caregivers of adolescents with T1DM experience a disease-related burden and elevated stress levels, which can influence the emotional and psychological well-being of their children [38]. A previous assessment of the caregivers who participated in this study revealed elevated levels of stress related to the frequency of caregiving tasks. However, their levels of strain associated with caregiving responsibilities were relatively low, suggesting that although caregivers perceive a high level of stress, they may not feel a corresponding burden. This finding underscores the intricate relationship between stress and perceived strain in caregiving contexts, thereby emphasizing the necessity for further investigation to elucidate the nuances of these phenomena.

In relation to the initial hypothesis, it was hypothesized that there would be positive associations between caregiver overload and emotional symptomatology, as well as the difficulties experienced by adolescents with T1DM. The existing literature suggests a relationship between the emotional state of caregivers and that of adolescents. For example, Evans et al. [15] identified a positive association between caregivers’ depressive symptoms and those of adolescents with T1DM, as well as observed that higher levels of diabetic distress in caregivers predict higher levels of distress in adolescents. This present study’s findings align with these observations, as it identified a positive and significant correlation between caregivers’ perceived stress and adolescents’ emotional distress. Specifically, this study found that caregiver stress levels were closely related to adolescents’ emotional distress and psychological adjustment. However, other recent studies [27,28,29] present conflicting results regarding the existence of emotional contagion in the dyad. For instance, Nguyen et al. [28] observed an absence of association between caregivers’ emotional distress and anxious–depressive symptomatology or blood glucose levels in their adolescent children with T1DM. However, this study’s limited sample size precludes the generalizability of its results, underscoring the necessity for additional research in this area.

In consideration of the second hypothesis, preliminary findings suggest a correlation between changes in caregiver and adolescent variables, with a decrease in means observed following the completion of the intervention program. These results indicate the potential for the program to have a beneficial impact on both caregivers and adolescents, potentially reducing perceived overload and enhancing their emotional well-being.

This present study’s findings align with those of previous research. Type 1 diabetes mellitus (T1DM) has been documented as a factor that can impact the family system in its entirety. Specifically, family discord related to diabetes, particularly psychological control, has been associated with a deterioration in the psychological state of adolescents, which can lead to the onset of depressive symptoms [39]. Conversely, effective communication and positive family functioning have been demonstrated to facilitate a balanced distribution of responsibilities related to disease management, thereby enhancing family cohesion and the adaptation of its members [21]. Luo et al. [24] underscored the pivotal role of family cohesion in enhancing resilience and self-management in adolescents with T1DM. These studies suggest that caregivers’ emotional states significantly influence adolescents’ ability to manage their illness, highlighting the importance of emotional contagion as a mechanism modulating this effect. This underscores the relevance of including caregivers in intervention programs targeting adolescents with T1DM.

The results of this study corroborate the notion of a reciprocal interaction between adolescents with T1DM and their caregivers, thereby providing empirical evidence that complements the extant scientific literature on the subject. The findings confirm the existence of emotional contagion within the dyad, suggesting that management of the disease and adherence to treatment may improve or deteriorate due to the mutual influence between caregivers’ perception of their ability to cope with T1DM and their children’s emotional state.

However, it is imperative to take the findings of this study within the limitations imposed by its design. The sample size was modest, and the absence of a control group limits the external validity of the results. To enhance the generalizability of the results, subsequent research endeavors should consider participant recruitment from diverse geographical locations. Furthermore, it would be beneficial to examine the relationship between psychological factors and medical outcomes using glycemic data to enhance the comprehension of the program’s effects. It is also imperative to consider potential confounding variables, such as socioeconomic background, family dynamics, or medical history, which could influence the observed relationships. To address this, it is recommended to employ multivariate methodologies that allow for more precise and detailed analysis, controlling for these variables and providing a more robust understanding of the interaction between the variables studied. Finally, further research is needed to assess the consequences of the intervention on psychological and clinical parameters, as well as to further explore the relationship between dyad members.

The results of this study corroborate the notion of a reciprocal interaction between adolescents with T1DM and their caregivers, thereby providing empirical evidence that complements the extant scientific literature on the subject. The findings confirm the existence of emotional contagion within the dyad, suggesting that the management of the disease and adherence to treatment may improve or deteriorate due to the mutual influence between caregivers’ perception of their ability to cope with T1DM and their children’s emotional state. However, for a more accurate interpretation of the results, it is essential to consider the presence of confounding variables, such as socioeconomic factors, family context, or clinical history of the participants, which could influence the observed relationships. In addition, the employment of multivariate methodologies that facilitate the simultaneous analysis of multiple interactions while accounting for these potential confounders would offer a more nuanced and reliable perspective on the dynamics between the variables under study.

In practical terms, the results of our study underscore the significance of family involvement in interventions targeting adolescents with type 1 diabetes mellitus (T1DM). The reciprocal influence of family members on disease management and coping is noteworthy. The dearth of studies in this area underscores the pressing need for further research, particularly in this domain, which is pivotal to enhancing the physical and socio–emotional well-being of adolescents and their families, thereby optimizing their quality of life.

In summary, the findings of this study underscore the need to design comprehensive interventions that address the medical needs of adolescents and provide psychological and educational support to caregivers. These interventions have the potential to generate important clinical benefits, such as increased adherence to treatment, improved glycemic control, and improvements in the socio–emotional well-being of both caregivers and adolescents. To further explore this field, researchers should adopt approaches that examine interactions within the caregiver–adolescent dyad and consider factors such as severity of illness, family environment, and access to supportive resources. These efforts are essential to developing more effective programs that optimize the quality of life for families coping with T1DM management.

However, this study has several limitations. The small sample size (15 dyads) and the absence of a control group limit the generalizability and external validity of the findings. Additionally, the nature of the intervention and the longitudinal design posed challenges for maintaining participant follow-up. Future research should address these issues by including larger, more diverse samples and integrating objective measures, such as glycemic outcomes, to better evaluate the program’s effects. Another significant limitation is the lack of confounder analysis or multivariate statistical approaches, which could provide deeper insights into the relationships between variables. Future studies should also investigate the intervention’s impact on both psychological and clinical parameters to build a more comprehensive understanding. These and other relevant considerations will be explored in subsequent research. The findings of this present study underscore the significance of integrating family members within interventions targeting adolescents diagnosed with type 1 diabetes mellitus (T1DM). The reciprocal influence that family members exert on the management of the disease and on their ability to cope with it is a salient factor that should not be overlooked in future therapeutic programs. The dearth of scientific literature on this subject underscores the pressing need for further research in this area, particularly through longitudinal studies comprising larger and more heterogeneous samples.

In practical terms, the findings of this study underscore the necessity of designing interventions that address not only the medical needs of adolescents but also include psychological and educational support strategies for caregivers. These interventions hold the potential to yield substantial clinical benefits, including enhanced adherence to treatment, improved glycemic control, and enhanced socio–emotional well-being for both adolescents and their families. To further explore this field, researchers are advised to prioritize integrative approaches that examine the dynamic interactions within the caregiver–adolescent dyad and consider the impact of factors such as disease severity, family environment, and access to supportive resources. These efforts are essential for the development of more effective programs that will optimize the quality of life for families coping with the management of T1DM.

## Figures and Tables

**Table 1 children-12-00151-t001:** 10Vida program design.

10VIDA PROGRAM (Sessions with Patients)		
Session	Theme	Aims
S1 Adolescent. My beliefs	Adjustment to illness	Assess, recognize, and value beliefs, concerns, or fears related to the disease
S2 Adolescent. A look inside me	Self-esteem/self-concept	To develop behavioral patterns that facilitate an adequate self-image and identity, without the stigmas of illness
S3 Adolescent. From serenity	Coping with fear	To learn to identify, attend to, and manage the anxious symptomatology associated with the life situations that a chronic disease in adolescence may entail. To favor a serene and positive attitude, knowing their own fears
S1 Parents. You take care of me; I take care of you	Caregiver needs	Know and address the psychological and emotional needs of primary caregivers by providing them with strategies. Intervene in the beliefs and concerns regarding the disease, their child, themselves, or the family
S4 Adolescent. Emotions: My friends	Emotional self-regulation	Encourage a coping and resilient attitude to facilitate the acquisition of appropriate habits and behaviors to promote positive emotions that can cushion the daily situations with the disease
S5 Adolescent. A look outside	Social area	To reflect on the importance of friendships at this age, and that they are sources of support in the face of illness and treatment
S2 Parents. Together, we make it happen	The family system	Emphasize the role of parents in coping with their child’s illness, reducing stress, and encouraging acceptance

**Table 2 children-12-00151-t002:** Clinical and sociodemographic data of the caregivers of adolescents with T1DM (Spain).

		n	%
Gender of caregiver	Male	0	0
	Female	15	100
Type of employment contract	Staff member	2	13.3
	Indefinite-term contract	5	33.3
	Others	8	53.3
Employment status of the caregiver	Yes	14	93.3
	No	1	6.7
Caregiver’s level of education	Uncompleted School Graduate	1	6.7
	School Graduate	2	13.3
	Baccalaureate/FP	4	26.7
	Higher education	8	53.3
Caregiver’s marital status	Married	10	66.7
	Separated	1	6.7
	Divorced	1	6.7
	Living as a couple	3	20.0
Relationship to adolescent	Mother	14	94.7
	Grandma	1	5.3
Physical problem of the caregiver	Yes	5	33.3
	No	10	66.7
Psychological problem of the caregiver	Yes	2	13.33
	No	13	86.67
Family socioeconomic level (annual income measured in euros)	High (EUR +100 thousand)	2	13.3
	High–Medium (EUR 45 to 99 thousand)	4	26.7
	Medium (EUR 25,500 to 44 thousand)	5	33.3
	Medium–Low (EUR 16 to 25 thousand)	2	13.3
	Low (EUR 10 thousand to 16 thousand)	2	13.3
Adolescent living with another family member with diabetes	Yes	2	10.5
	No	17	89.5

Note. % = Percentage, n = number of cases.

**Table 3 children-12-00151-t003:** Clinical and sociodemographic data of the adolescents with T1DM (Spain).

		n	%
Adolescent Gender	Male	9	60.0
	Female	6	40.0
Disease/Disorder Secondary/Adolescent	None	9	60.0
	Physical	4	26.7
	Psychological	2	13.3
Use of adrenaline	Yes	1	6.7
	No	14	93.3
Frequency of specialist visits	Every 3 months	11	73.3
	Every 4 months	2	13.3
	Every 6 months	1	6.7
	Annual	1	6.7

Note. % = Percentage, n = number of cases.

**Table 4 children-12-00151-t004:** Quantitative sociodemographic data about caregivers and adolescents (Spain).

	Mean	Median	SD	Minimum	Maximum
Adolescent age	13.46	13	1.68	11.00	17.00
Age of caregiver	47.66	47	7.21	35.00	65.00
Time in treatment	45.26	33	40.24	6.00	138.00
Number of hospital admissions	1.60	1.00	1.639	0	6
Duration of admission (adolescent)	1.20	0.00	1.740	0	5
Months active, employment (caregiver)	142.31	132.00	97.713	12	276

SD = Standard Deviation.

**Table 5 children-12-00151-t005:** Profile of pre- and post-intervention variables of caregivers and adolescents with T1DM (Spain).

		Mean	Median	SD	Minimum	Maximum
Pre-intervention scores	Perceived Stress due to Caregiving Tasks Frequency: Communication (PIP)	9.93	9.00	1.94	7.00	13.00
	Perceived Stress due to Caregiving Tasks Effort: Communication (PIP)	6.00	5.00	3.74	3.00	15.00
	Perceived Stress due to Caregiving Tasks Frequency: Medical Care (PIP)	13.00	14.00	2.20	9.00	15.00
	Perceived Stress due to Caregiving Tasks Effort: Medical Care (PIP)	6.87	4.00	4.55	3.00	15.00
	Perceived Stress due to Caregiving Tasks Frequency: Emotional Distress (PIP)	10.07	10.00	3.58	5.00	15.00
	Perceived Stress due to Caregiving Tasks Effort: Emotional Distress (PIP)	9.20	9.00	3.45	4.00	15.00
	Perceived Stress due to Frequency of Caregiving Tasks: Family Role (PIP)	7.73	7.00	3.05	4.00	15.00
	Perceived due to Caregiving Tasks Effort: Family Role (PIP)	7.33	6.00	3.39	3.00	15.00
	Perceived Stress due to Frequency of Caregiving Tasks: General (PIP)	10.18	9.75	2.15	7.00	14.50
	Perceived stress due to Caregiving Tasks Effort: General (PIP)	7.35	5.75	3.20	3.75	13.50
	General Adolescent Emotional Distress (HADS)	6.20	5.00	5.45	1.00	20.00
	General Adolescent Emotional Adjustment (SDQ)	10.13	9.00	6.84	1.00	29.00
Post-intervention scores	Perceived Stress due to Caregiving Tasks Frequency: Communication (PIP)	9.47	10.00	1.96	5.00	13.00
	Perceived Stress due to Caregiving Tasks Effort: Communication (PIP)	4.73	3.00	2.89	3.00	11.00
	Perceived Stress due to Caregiving Tasks Frequency: Medical Care (PIP)	12.73	13.00	2.28	8.00	15.00
	Perceived Stress due to Caregiving Tasks Effort: Medical Care (PIP)	5.60	3.00	4.10	3.00	14.00
	Perceived Stress due to Caregiving Tasks Frequency: Emotional Distress (PIP)	8.53	7.00	3.78	4.00	15.00
	Perceived Stress due to Caregiving Tasks Effort: Emotional Distress (PIP)	7.40	7.00	3.44	3.00	14.00
	Perceived Stress due to Frequency of Caregiving Tasks: Family Role (PIP)	7.13	7.00	2.85	4.00	13.00
	Perceived due to Caregiving Tasks Effort: Family Role (PIP)	6.53	6.00	3.15	3.00	15.00
	Perceived Stress due to Frequency of Caregiving Tasks: General (PIP)	9.46	9.25	2.37	6.25	13.50
	Perceived stress due to Caregiving Tasks Effort: General (PIP)	6.06	4.75	3.10	3.50	13.25
	General Adolescent Emotional Distress (HADS)	5.93	4.00	4.89	.00	14.00
	General Adolescent Emotional Adjustment (SDQ)	9.66	9.00	7.20	1.00	25.00
Difference between pre–post scores	Perceived Stress due to Caregiving Tasks Frequency: Communication (PIP)	0.4667	1.00	1.40	−2.00	3.00
	Perceived Stress due to Caregiving Tasks Effort: Communication (PIP)	1.26	0	3.73	−5.00	12
	Perceived Stress due to Caregiving Tasks Frequency: Medical Care (PIP)	0.26	1.00	2.12	−5.00	4.00
	Perceived Stress due to Caregiving Tasks Effort: Medical Care (PIP)	1.26	0	3.05	−1.00	11.00
	Perceived Stress due to Caregiving Tasks Frequency: Emotional Distress (PIP)	1.53	1.00	2.23	−4.00	5.00
	Perceived Stress due to Caregiving Tasks Effort: Emotional Distress (PIP))	1.80	1.00	1.65	−1.00	5.00
	Perceived Stress due to Frequency of Caregiving Tasks: Family Role (PIP)	0.60	1.00	2.47	−6.00	3.00
	Perceived due to Caregiving Tasks Effort: Family Role (PIP)	0.80	1.00	2.007	−3.00	4.00
	Perceived Stress due to Frequency of Caregiving Tasks: General (PIP)	0.71	1.00	1.37	−2.25	2.25
	Perceived stress due to Caregiving Tasks Effort: General (PIP)	1.28	0.75	1.87	−1.75	5.75
	General Adolescent Emotional Distress (HADS)	0.26	1.00	5.24	−13.00	8.00
	General Adolescent Emotional Adjustment (SDQ)	0.46	1.00	4.51	−7.00	8.00

SD = Standard Deviation.

**Table 6 children-12-00151-t006:** Correlations between the variation of the main variables of caregivers and adolescents (Spain).

	1	2	3	4	5	6	7	8	9	10	11	12
1. Variation of Stress due to Caregiving Tasks Frequency: Communication (PIP)	1											
2. Variation of Stress due to Caregiving Tasks Effort: Communication (PIP)	0.072	1										
3. Variation of Stress due to Caregiving Tasks Frequency: Medical Care (PIP)	0.039	0.219	1									
4. Variation of Stress due to Caregiving Tasks Effort: Medical Care (PIP)	0.053	0.759 **	0.022	1								
5. Variation of Stress due to Caregiving Tasks Frequency: Family Role (PIP)	0.296	0.087	−0.201	−0.003	1							
6. Variation of Stress due to Caregiving Tasks Effort: Family Role (PIP)	0.329	0.301	−0.110	0.383	0.643 **	1						
7. Variation of Stress due to Caregiving Tasks Frequency: Emotional Distress (PIP)	0.110	0.092	−0.256	0.054	0.839 **	0.625 *	1					
8. Variation of Stress due to Caregiving Tasks Effort: Emotional Distress (PIP))	0.079	0.243	−0.437	0.186	0.259	0.252	0.315	1				
9. Variation of Stress due to Caregiving Tasks Frequency: General Score	0.521 *	0.337	0.126	0.201	0.829 **	0.731 **	0.755 **	0.092	1			
10. Variation of Stress due to Caregiving Tasks Effort: General Score	0.208	0.776 **	−0.079	0.809 **	0.210	0.544 *	0.189	0.594 *	0.298	1		
11. Variation of General Adolescent Emotional Distress (HADS)	0.264	0.334	−0.088	0.017	0.721 **	0.498	0.602 *	0.173	0.740 **	0.157	1	
12. Variation of General adolescent emotional adjustment (SDQ)	0.262	0.255	0.126	−0.118	0.572 *	0.203	0.513	0.370	0.614 *	0.069	0.722 **	1

* *p* < 0.05, ** *p* < 0.01.

## Data Availability

The data presented in this study are available on request from the corresponding author. They are not publicly available due to data privacy and confidentiality.

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
