# Peer review of "Emotional Contagion Among Adolescents with Type 1 Diabetes Mellitus (T1DM) and Their Primary Caregivers: Benefits of Psychological Support for Family Systems in Pilot Study"

_children, 2025, doi:10.3390/children12020151_

Round 1
Reviewer 1 Report
Comments and Suggestions for Authors
Please respect the journal's requirements regarding numbering of author affiliations.
You need to delete lines 47-55 in the Introduction section.
The Introduction and the Discussion parts can be further improved by eferring/comparison to existing literature.
Authors should indicate during which period they conducted the study.
How was the sample size calculated? (important for statistical validity of the results of the study)
The experience of those conducting the questionnaire should be stated. Who are the members of the therapeutic team? What kind of psychotherapy did you used?
Please add some comments in the limitation because there is no appropriate control (for confounders) and no multivariate analysis in this study.
There should be future suggestions / take-home lesson related to your study for the investigators who want to continue this kind of research. Also mention practical clinical benefits of the results of this study.
Author Response
Dear Ms. Adriana Li and reviewers whom we would like to thank for the effort and time invested in reviewing our manuscript. We have tried to include the suggestions discussed and presented below, and we hope that, once made, we will be able to continue with the journal publication process. We appreciate the time and effort invested in the review because we feel that the manuscript has been greatly improved. Below is the response to each of the suggestions made, the changes made have been marked with change control.
Reviewer 1
Comment: Please respect the journal's requirements regarding numbering of author affiliations.
Response: Following the suggestions, we have corrected the numbering indicating the affiliations of all authors.
Comment: You need to delete lines 47-55 in the Introduction section.
Response: Thank you for the indication. The error in question has been rectified by the deletion of the previously mentioned lines.
Comment: The Introduction and the Discussion parts can be further improved by referring/comparison to existing literature.
Response: We appreciate your suggestion and would like to inform you that we have expanded and modified the sections indicated. This modification included the addition of more references and comparisons with the existing scientific literature. A growing body of research has identified the relationship between caregivers and adolescents as a significant area of focus, emphasizing the role of emotional contagion in this dynamic interaction.
Comment: Authors should indicate during which period they conducted the study.
Response: As recommended, the collection of the sample was incorporated into the methodology and participants section.
Comment: How was the sample size calculated? (important for statistical validity of the results of the study)
Response: Thanks for the comment. This paper presents the preliminary findings of a pilot study, the sample of which was derived from all patients who met the inclusion criteria for the Paediatric Endocrinology Unit at the Hospital General Universitario de Valencia, a reference centre in the province for paediatric hospital care in T1DM. It is important to note that, given the longitudinal nature of this study, which involves the analysis of emotional contagion between parents and adolescents only after the completion of all sessions by both parties, a portion of the initial sample was unfortunately lost due to experimental mortality, i.e. due to either the parents or the adolescents not attending all sessions. Consequently, of the initial pilot sample, a mere 60% could be included in this study.
Comment: The experience of those conducting the questionnaire should be stated. Who are the members of the therapeutic team? What kind of psychotherapy did you used?
Response: We have added this information in the ‘Design and Procedure’ section, thank you very much for letting us know.
Comment: Please add some comments in the limitation because there is no appropriate control (for confounders) and no multivariate analysis in this study
Response: Thanks for pointing that out, we have added this issue to the “limitations” section.
Comment: There should be future suggestions / take-home lesson related to your study for the investigators who want to continue this kind of research. Also mention practical clinical benefits of the results of this study.
Response: We would like to express our gratitude for your insightful comments. In response to the recommendations provided, we have expanded the scope to encompass a more comprehensive array of practical implications and valuable insights.
Reviewer 2 Report
Comments and Suggestions for Authors
Abstract
The authors made the following statement:
‘ To this end, variables were created whose value was the difference between the pre- and post-treatment scores.’
It is not clear from this context what is meant by scores. What is being measured and scored here? Please elaborate.
Line 27: what is PIP? Please spell out abbreviations accordingly
Please spell out all abbreviations upon its first appearance in the text such as HADS and SDQ
Please declare the study design in the abstract. Is this an interventional study?
Introductions
The authors made the following paragraph:
‘The introduction should briefly place the study in a broad context and highlight why it is important. It should define the purpose of the work and its significance. The current state of the research field should be carefully reviewed and key publications cited. Please highlight controversial and diverging hypotheses when necessary. Finally, briefly mention the main aim of the work and highlight the principal conclusions. As far as possible, please keep the introduction comprehensible to scientists outside your particular field of research. References should be numbered in order of appearance and indicated by a numeral or numerals in square brackets—e.g., [1] or [2,3], or [4–6]. See the end of the document for further details on references.’
This does not make sense. Please revise.
Line 66: T1DMD. This is unusual abbreviation for type 1 diabetes .
Lines 84 - 86 : The authors made the following statement:
‘In this sense, the 10Vida treatment program was developed to intervene at the psychosocioemotional and educational level in adolescents with T1DM and their families, providing them with tools that facilitate the proper adjustment and management of this disease.’
Developed by who? When? In what context? Please clarify.
The developed hypothesis added by the end of the introduction is not clear and does not seem to generate clear research objectives. Please revise.
Methods
Line 100:
‘The final sample comprised 15 primary family-caregiver-adolescent dyads’
What is meant by the final sample? Was there a primary sample? If yes, please elaborate on the number or excluded cases and exclusion criteria.
Line 114: what is the table being referred to in this sentence? Please add the correct number of the table.
Lines 244- 225:
‘In order to respond to the hypotheses mentioned, it was necessary to perform an analysis of the most relevant descriptive statistics (mean, median, standard deviations, etc.).’
To enhance the scientific transparency, please declare all statistical tests applied in the current analysis. Using ‘etc’ to indicate tests that were not declared is misleading.
Lines 226- 227: Likewise, Spearman correlations were used to check the relationships produced between the study variables, since some of the variables did not meet the assumptions of normality.
Please declare all variables used in the correlation analysis.
What is the designated statistical significance level for the current analysis? was it 0.05?
Tables: please elaborate on the title of the tables to indicate where the data was collected and type of the population.
What is the currency of the income?
Tables 4 and 5: what is meant by media? What is meant by DT?
Table 5 should be redesigned to summarise the difference between the pre and post assessments. Use of visualization is also advised to display the difference levels.
Table 6 can be combined with table 5
Table 7 is not clear. Please redesign.
Discussion
Lines 316 – 334: the first three paragraph seems very similar to the content of the introduction. Please revise to focus more on the study findings.
Discussion is very limited and does not properly compare the findings to the similar literature.
The discussion should include a section indicating strengths and weaknesses of the current study.
Conclusion
The produced conclusions and recommendations are largely affected by the ambiguity of the data analysis. The design is not based on clear sample size estimation. Differences between the pre- and post scores are not properly tested via the relevant tests that are suitable for the data distribution. Provision of statistical significance levels is very limited. Additionally, no control for confounding factors was provided at all.
Comments on the Quality of English Language
The manuscript can benefit from English language editing service
Author Response
Dear Ms. Adriana Li and reviewers whom we would like to thank for the effort and time invested in reviewing our manuscript. We have tried to include the suggestions discussed and presented below, and we hope that, once made, we will be able to continue with the journal publication process. We appreciate the time and effort invested in the review because we feel that the manuscript has been greatly improved. Below is the response to each of the suggestions made, the changes made have been marked with change control.
Reviewer 2
Comment: The authors made the following statement: ‘To this end, variables were created whose value was the difference between the pre- and post-treatment scores.’ It is not clear from this context what is meant by scores. What is being measured and scored here? Please elaborate.
Response: Thanks for the comment. This information is not explained in depth in this section due to word count requirements. Nevertheless, we have changed the way that we describe this variable in the abstract section to clarify this matter. The full explanation of the process can be found in section 3.2.3.
Comment: Line 27: what is PIP? Please spell out abbreviations accordingly. Please spell out all abbreviations upon its first appearance in the text such as HADS and SDQ
Response: Thank you for your observation. The issue was similar to the one mentioned earlier—we had only included the abbreviation ‘PIP’ in the abstract due to word limit constraints. However, we would like to clarify that the abbreviations for all questionnaires (such as HADS and SDQ) are explained in detail in section 2.3.2.1. Additionally, to address this concern, we have removed the abbreviations from the abstract to enhance clarity.
Comment: Please declare the study design in the abstract. Is this an interventional study?
Response: Thanks for pointing it out. The study constitutes a correlational analysis, in which the relationship between the indicators that have been generated is analysed in order to study the change that occurs before and after the administration of the treatment.
Comment: The authors made the following paragraph: ‘The introduction should briefly place the study in a broad context and highlight why it is important. It should define the purpose of the work and its significance. The current state of the research field should be carefully reviewed and key publications cited. Please highlight controversial and diverging hypotheses when necessary. Finally, briefly mention the main aim of the work and highlight the principal conclusions. As far as possible, please keep the introduction comprehensible to scientists outside your particular field of research. References should be numbered in order of appearance and indicated by a numeral or numerals in square brackets—e.g., [1] or [2,3], or [4–6]. See the end of the document for further details on references.’ This does not make sense. Please revise.
Response: Thank you for the indication. The error in question has been rectified by the deletion of the previously mentioned lines.
Comment Line 66: T1DMD. This is unusual abbreviation for type 1 diabetes.
Response: We appreciate your notification. We have already corrected the errors in the nomenclature of the abbreviations, unifying them all with T1DM.
Comment: Lines 84 - 86 : The authors made the following statement: ‘In this sense, the 10Vida treatment program was developed to intervene at the psychosocioemotional and educational level in adolescents with T1DM and their families, providing them with tools that facilitate the proper adjustment and management of this disease.’ Developed by who? When? In what context? Please clarify.
Response: We appreciate your suggestion and would like to inform you that we have added more information regarding the program in order to clarify the content.
Comment: The developed hypothesis added by the end of the introduction is not clear and does not seem to generate clear research objectives. Please revise.
Response: Thank you for your suggestion. We have restated the hypotheses to enhance clarity. Comment: Line 100: ‘The final sample comprised 15 primary family-caregiver-adolescent dyads’ What is meant by the final sample? Was there a primary sample? If yes, please elaborate on the number of excluded cases and exclusion criteria.
Response: Thanks for the remark, as we pointed out earlier, this was a pilot study, and we could only include 60% of the initial sample.
Comment: Line 114: what is the table being referred to in this sentence? Please add the correct number of the table.
Response: As indicated above, the table reference has been added.
Comment: Lines 244- 225: ‘In order to respond to the hypotheses mentioned, it was necessary to perform an analysis of the most relevant descriptive statistics (mean, median, standard deviations, etc.).’ To enhance the scientific transparency, please declare all statistical tests applied in the current analysis. Using ‘etc’ to indicate tests that were not declared is misleading.
Response: Thank you for pointing this out and apologies for the error. We specified the analysis made in the study and removed “etc.”.
Comment: Lines 226- 227: Likewise, Spearman correlations were used to check the relationships produced between the study variables, since some of the variables did not meet the assumptions of normality. Please declare all variables used in the correlation analysis.
Response: In accordance with the recommendations provided by the reviewer, the nature of the variables incorporated within the scope of this study has been delineated. Furthermore, the integration of all subscales from the Perceived Stress Questionnaire has been implemented to enhance the conciseness of the results.
Comment: What is the designated statistical significance level for the current analysis? Was it 0.05?
Response: The designated statistical significance level for the correlations can be found below Table 6. The significant correlations are marked with an asterisk following this rule: *= p<0.05, ** p<0.01.
Comment: Tables: please elaborate on the title of the tables to indicate where the data was collected and type of the population.
Response: Table titles have been revised and the country of collection has been included.
Comment: What is the currency of the income?
Response: Thanks for pointing that out, the currency is measured in Euros. We have added the information to table 2.
Comment: Tables 4 and 5: what is meant by media? What is meant by DT?
Response: Thanks for pointing that out, it was a translation issue. Media was “Mean” and “DT” was “Standard deviation”. It has been corrected.
Comment: Table 5 should be redesigned to summarise the difference between the pre and post assessments. Use of visualization is also advised to display the difference levels. Table 6 can be combined with table 5
Response: Thanks for the recommendation, we have decided to merge Tables 5 and 6. Also, we have added a figure to represent the difference levels between the variables
Comment: Table 7 is not clear. Please redesign.
Response: Thanks for the comment. We have revised the way we described the variables in the correlation table to make sure it is clearer. The design of the table is the standard way of depicting a correlation.
Comment: Lines 316 – 334: the first three paragraph seems very similar to the content of the introduction. Please revise to focus more on the study findings.
Response: Thank you for your indications. We have revised and reworked the first paragraphs of the discussion to address the points you raised.
Comment: Discussion is very limited and does not properly compare the findings to the similar literature. The discussion should include a section indicating strengths and weaknesses of the current study
Response: Thank you for your indications. We have revised the mentioned sections and added information in order to address the points you raised.
Comment: Conclusion. The produced conclusions and recommendations are largely affected by the ambiguity of the data analysis. The design is not based on clear sample size estimation. Differences between the pre- and post scores are not properly tested via the relevant tests that are suitable for the data distribution. Provision of statistical significance levels is very limited. Additionally, no control for confounding factors was provided at all. The manuscript can benefit from English language editing service.
Response: We understand your concerns and as we have indicated this is a pilot and preliminary study on the study of the dyad between caregivers and adolescents with type 1 diabetes mellitus. We know that the results cannot be generalized but it is a first attempt to try to clarify the relationship between these variables. The English has also been revised to try to make it more fluent.
Round 2
Reviewer 1 Report
Comments and Suggestions for Authors
I have no other suggestions.
Author Response
Dear Ms. Adriana Li and reviewers whom we would like to thank for the effort and time invested in reviewing our manuscript. We have tried to include the suggestions discussed and presented below, and we hope that, once made, we will be able to continue with the journal publication process. We appreciate the time and effort invested in the review because we feel that the manuscript has been greatly improved. Below is the response to each of the suggestions made, the changes made have been marked with change control.
Reviewer 1
Comment: I have no other suggestions.
Response: Thank you for your effort to revise the manuscript.
Reviewer 2 Report
Comments and Suggestions for Authors
I wish to thank the authors for responding to my comments. The quality of the manuscript has improved. However, please note the following:
It should be declared early in the title and abstract of the paper that this study is a pilot study
Secondly, please add the figure indicated in one of the responses. I was not able to find it.
Best of Luck
Author Response
Dear Ms. Adriana Li and reviewers whom we would like to thank for the effort and time invested in reviewing our manuscript. We have tried to include the suggestions discussed and presented below, and we hope that, once made, we will be able to continue with the journal publication process. We appreciate the time and effort invested in the review because we feel that the manuscript has been greatly improved. Below is the response to each of the suggestions made, the changes made have been marked with change control.
Comment: I wish to thank the authors for responding to my comments. The quality of the manuscript has improved. However, please note the following:
Response: Thank you very much for your comments and the effort made in improving the manuscript and revising it.
Comment: It should be declared early in the title and abstract of the paper that this study is a pilot study
Response: Following suggestions, the following has been incorporated in the title and in the abstract
Comment: Secondly, please add the figure indicated in one of the responses. I was not able to find it.
Response: We are sorry for the confusion at the beginning we were going to incorporate the figure but when unifying the information of the tables following the indicated we think that it is enough to clarify the information.